# Nutritional Composition and Bioactive Compounds in Tomatoes and Their Impact on Human Health and Disease: A Review

**DOI:** 10.3390/foods10010045

**Published:** 2020-12-26

**Authors:** Md Yousuf Ali, Abu Ali Ibn Sina, Shahad Saif Khandker, Lutfun Neesa, E. M. Tanvir, Alamgir Kabir, Md Ibrahim Khalil, Siew Hua Gan

**Affiliations:** 1Laboratory of Preventive and Integrative Biomedicine, Department of Biochemistry and Molecular Biology, Jahangirnagar University, Savar, Dhaka 1342, Bangladesh; yousufbmb40ju@gmail.com (M.Y.A.); shahadsaifkhandker@gmail.com (S.S.K.); alamgirju42@gmail.com (A.K.); 2Department of Biochemistry and Molecular Biology, Gono Bishwabidyalay, Savar, Dhaka 1344, Bangladesh; 3Center for Personalized Nanomedicine, Australian Institute for Bioengineering and Nanotechnology (AIBN), The University of Queensland, Brisbane, QLD 4072, Australia; a.sina@uq.edu.au; 4Department of Biotechnology and Genetic Engineering, Bangabandhu Sheikh Mujibur Rahman Science and Technology University, Gopalganj, Dhaka 8100, Bangladesh; l.neesa@bgeju.edu.bd; 5Veterinary Drug Residue Analysis Division, Institute of Food and Radiation Biology, Atomic Energy Research Establishment, Savar, Dhaka 1349, Bangladesh; tanvir.ju.37@gmail.com; 6School of Pharmacy, Pharmacy Australia Centre of Excellence, The University of Queensland, Woolloongabba, QLD 4102, Australia; 7School of Pharmacy, Monash University Malaysia, Jalan Lagoon Selatan, Bandar Sunway, Selangor 47500, Malaysia

**Keywords:** nutrients, tomatoes, phytochemicals, antioxidants, human health, degenerative diseases

## Abstract

Tomatoes are consumed worldwide as fresh vegetables because of their high contents of essential nutrients and antioxidant-rich phytochemicals. Tomatoes contain minerals, vitamins, proteins, essential amino acids (leucine, threonine, valine, histidine, lysine, arginine), monounsaturated fatty acids (linoleic and linolenic acids), carotenoids (lycopene and β-carotenoids) and phytosterols (β-sitosterol, campesterol and stigmasterol). Lycopene is the main dietary carotenoid in tomato and tomato-based food products and lycopene consumption by humans has been reported to protect against cancer, cardiovascular diseases, cognitive function and osteoporosis. Among the phenolic compounds present in tomato, quercetin, kaempferol, naringenin, caffeic acid and lutein are the most common. Many of these compounds have antioxidant activities and are effective in protecting the human body against various oxidative stress-related diseases. Dietary tomatoes increase the body’s level of antioxidants, trapping reactive oxygen species and reducing oxidative damage to important biomolecules such as membrane lipids, enzymatic proteins and DNA, thereby ameliorating oxidative stress. We reviewed the nutritional and phytochemical compositions of tomatoes. In addition, the impacts of the constituents on human health, particularly in ameliorating some degenerative diseases, are also discussed.

## 1. Introduction

Tomatoes (*Solanum lycopersicum* L.), which are frequently included in the Mediterranean diet and are widely consumed as vegetables, play an important role in nutrition because of their well-established health benefits [1]. Tomatoes are used in many processed food products such as sauces, salads, soups, and pastes [2]. Common nutrients reported to be present in tomatoes are vitamins, minerals, fiber, protein, essential amino acids, monounsaturated fatty acids, carotenoids and phytosterols [3,4,5,6]. These nutrients perform various body functions including constipation prevention, reduction in high blood pressure, stimulation of blood circulation, maintenance of lipid profile and body fluids, detoxification of body toxins and maintaining bone structure as well as strength [1,7,8]. Tomatoes are also an excellent source of nutrients and bioactive compounds, commonly known as secondary metabolites, the concentrations of which are correlated with the prevention of human chronic degenerative diseases, such as cardiovascular disease (CVD), cancer, and neurodegenerative diseases [9,10,11]. Due to the high concentrations of different natural antioxidant chemicals, such as carotenoids (β-carotenoids and lycopene), ascorbic acid (vitamin C), tocopherol (vitamin E) and bioactive phenolic compounds (quercetin, kaempferol, naringenin and lutein, as well as caffeic, ferulic and chlorogenic acids), tomatoes can help ameliorate many diseases, especially chronic diseases [12,13]. These compounds play beneficial roles in inhibiting reactive oxygen species (ROS) by scavenging free radicals, inhibiting cellular proliferation and damage, inhibiting apoptosis as well as metal chelation, modulation of enzymatic activities, cytokine expression and signal transduction pathways [12,14]. The main carotenoid in tomato is lycopene, which is responsible for its red color. The pharmacological activities of lycopene and other phenolic compounds include anticancer, anti-inflammatory, antidiabetic, anti-allergenic, anti-atherogenic, antithrombotic, antimicrobial, antioxidant, vasodilator and cardioprotective effects [15,16,17,18]. In addition to having good nutritive value and health promoting activities, the polyphenolic compounds and carotenoids also contribute to sensory activities including maintaining good aroma, taste, and texture [19]. Tomato is an important dietary source of both soluble and insoluble dietary fibers, namely cellulose, hemicelluloses and pectins [20]. In general, these fibers are resistant to intestinal digestion in the large intestine and are believed to ameliorate bowel disorders, cancer, diabetes, CVDs, and obesity [21,22]. Important proximate composition parameters for tomatoes include sugar content, pH, energy, acidity and reducing sugar contents [4]. The proximate compositions help in the characterization and identification of tomato nutrients. The combination of vitamins, minerals, amino acids, and fats all together contribute to making tomato part of a balanced diet. Phytosterols, which are involved in the prevention of colon cancer and heart disease, are present in tomatoes in lower amounts than that found in other fruits and vegetables [23]. Among the phytosterols, β-sitosterol, campesterol and stigmasterol are the main ones [5]. The antioxidant compounds predominately present in tomato consist of several different types of carotenoids, vitamin C, vitamin E, and phenolic compounds that confer their antioxidant activities by neutralizing reactive oxygen species (ROS) and protecting the cell membrane against lipid peroxidation [24,25].

Nutritional composition of tomato varies based on the tomato cultivar, extraction procedures, analysis methods and environmental conditions. During the processing of tomato products, up to 30% of their original weight are turned into waste, which may still contain some nutritive values [26]. For example, the seeds and the peel are the main waste product of tomato, which are rich in protein, dietary fibers, bioactive compounds and lycopene [27]. The by-products are used as food additives especially in the meat industries [28]. Nevertheless, although the waste products of tomato are a rich source of nutrients, proper research should be undertaken before their consumption. In spite of having health benefits, tomatoes demonstrate some undesired effects on the body when consumed in large amounts or in abnormal body conditions. The adverse effects of tomato intake are associated with renal problems, allergies, arthritis, heartburn, and migraine [1].

Although scattered data are available, there is a lack of updated compiled information on the nutrient composition and the health benefits of tomatoes. Therefore, in this review, we bring together information on all the nutrient compositions of tomato such as proximate composition, minerals, heavy metals, vitamins, fatty acids, amino acids, carotenoids, phytosterols, antioxidant activity and different types of bioactive compounds. Consequently, we discuss the associated health benefits of the bioactive compounds present in tomato in preventing chronic degenerative diseases, such as CVDs, diabetes, and cancer.

## 2. Methodology

A comprehensive literature search was performed by combining the appropriate keywords including “tomato”, “nutritional composition”, “proximate composition”, “phytochemicals”, “physiochemical properties”, “mineral”, “vitamin”, “fatty acid”, “amino acid”, “carotenoid”, “phytosterol”, “antioxidant properties”, “bioactive compounds”, “health benefits”, “human degenerative diseases”, “cardiovascular diseases”, “diabetes” and “cancer”. As for the search engine, Google Scholar, Scopus, Web of Science, ScienceDirect, and Pubmed were independently searched. Only English language published articles were considered. There was no year restriction, and the final search was conducted on 27 July 2020. The references were managed with EndNote software (version X7).

## 3. Nutritional Composition of Tomato

### 3.1. Proximate Composition

Proximate analysis is one of the first approaches for food characterization, particularly for the identification of nutrients in any food products. Generally, water, ash, protein, lipid, carbohydrate, sugar and reducing sugar contents, as well as pH, energy and acidity are the key proximate compositions of a food sample [29]. For instance, ash content is an important step in the analysis of nutritional element contents in food products. Ash refers to the inorganic residue (mineral content) that remains after the complete oxidation of organic matter and removal of water by heating (ashing) of a food sample in a furnace [30,31]. Next, moisture content (total solids) is important because it affects the chemical and physical aspects of food, which determine its freshness and storage stability [32,33]. Protein, lipids, and carbohydrates are principal components of foods and are the main elements in proximate composition analysis.

Proteins, which are macromolecules present in food, are important for cellular structure and biological functions. Protein analysis is crucial for nutritional labeling, as well as in describing the biological activities and functional properties of food products [33,34,35]. Lipids are another group of macromolecules that are generally insoluble in water but are soluble in organic solvents. In fact, precise and accurate analysis of lipid content in food is mandatory for the standard of quality and nutritional labeling and is important in ensuring manufacturing specification [36].

Carbohydrate analysis is also important as a major (more than 70%) energy source. Carbohydrate analysis yields nutritional information, standard of identity, water holding capacity, flavors, desirable textures, and stability of food products [37,38]. In addition, pH analysis of food samples is essential for food processing and storage. Dietary fiber is another important component of proximate analysis because it ensures a variety of health benefits, including protection against heart disease, colon cancer and diabetes [39]. In a more recent study based on previously published original research articles, an average tomato consists of ash 0.6%, water 94.7 (g/100 g), moisture 93.76 (g/100 g), total protein 0.56 (g/100 g), lipid 0.13 (g/100 g), carbohydrates 5.96 (g/100 g), total sugar 4.41 (g/100 g), pH 4.38, energy 26.60 kcal/100 g, acidity 0.48%, reducing sugar 35.84%, fructose 2.88%, glucose 2.45%, sucrose 0.02% and total fiber 1.54 (g/100 g) (Table 1).

### 3.2. Mineral Content

Minerals are naturally occurring inorganic solid substances. They are essential for a variety of bodily functions, including the regulation of metabolic pathways, formation of vital organs, maintenance of bodily physiological functions, regulation of pH balance, fluid balance, blood pressure, nerve transmission, muscle contraction and energy production [53,54,55,56]. Some minerals, such as calcium (Ca), potassium (K), sodium (Na), phosphorus (P), magnesium (Mg), sulfur (S) and chlorine (Cl), are highly essential (average daily intake ˃ 50 mg) and are therefore known as major elements. Others include iron (Fe), iodine (I), zinc (Zn), fluorine (F), copper (Cu), selenium (Se), manganese (Mn), cobalt (Co), chromium (Cr), nickel (Ni), molybdenum (Mo) and selenium (Se), which are required in comparatively smaller amounts (< 50 mg/day) and are known as trace elements. Other elements, such as aluminum (Al), arsenic (As), boron (B), barium (Ba), bismuth (Bi), bromine (Br), lead (Pb), cadmium (Cd), cesium (Cs), germanium (Ge), lithium (Li), mercury (Hg), rubidium (Rb), silicon (Si), antimony (Sb), tin (Sn), samarium (Sm), strontium (Sr), tungsten (W), titanium (Ti) and thallium (Tl), which are needed in even smaller amounts, (1 µg/day) are known as ultratrace elements [57,58,59]. Pb, As, Hg, Cd, Cu, Cr, Ni, Zn and Mn are heavy metals that are toxic if present in low concentrations because they tend to accumulate in living cells [59,60].

From a nutritional perspective, tomato is a good source of minerals and other elements [4,44]. In this review, 23 types of minerals and their amounts present in tomato peel fiber are compiled, including the major elements (calcium, potassium, sodium, phosphorus, magnesium, sulfur, chlorine) and trace elements (iron, iodine, zinc, fluorine, cupper, manganese, cobalt, chromium, nickel, aluminum, arsenic, boron, lead, cadmium, nitrate, chlorine, selenium, silicon) (Table 2).

### 3.3. Vitamin Content

The accurate and precise analysis of vitamin content is important for a standard balanced diet because low or excessive amounts of vitamins can contribute to disease conditions by hampering normal cell growth [64]. Tomatoes are one of the most versatile and widely consumed vegetables in many countries and are a rich source of vitamins [65,66]. Vitamins C, B-complex, A, E and K are the main types of vitamins present in tomato, with vitamin C reported to be the highest (Table 3). Vitamins C and E (tocopherol) exhibit antioxidant activities making tomato a useful therapeutic agent for the prevention of various diseases, including CVDs and cancer [12,67,68,69,70]. Among the various types of vitamin B-complexes, the amount of folate is comparatively high in tomatoes. Nevertheless, excessive amounts of water-soluble vitamin B do not cause any toxicity because these vitamins can be easily excreted from the body. Vitamins perform various functions, such as maintaining the nervous system, producing red blood cells and enzymatic function [71,72].

### 3.4. Fatty Acid Content

Tomato contains many different types of fatty acids (Table 4). Among them, linoleic and polyunsaturated fatty acids are the highest. Linoleic and linolenic acids are two essential fatty acids. Since the essential fatty acids cannot be synthesized by humans or animals, they must come from the dietary sources and tomato provides a good source of these acids. On the other hand, polyunsaturated fatty acids are also very important for the body since they are essential for the maintenance of plasma membrane integrity, cell growth and prevention of disease [77,78]. From this point of view, tomato is therefore a rich and highly nutritious food product.

### 3.5. Amino Acid Content

Amino acids are the building blocks of proteins that conduct important bodily functions, including the maintenance of cellular structure, transport and storage of nutrients, wound healing, and repair of damaged tissues [79]. A total of 17 amino acids have been identified in tomato (Table 5). It is estimated that essential amino acids constitute 39.75% of the total protein in tomato. Among these, the highest was glutamic acid (approximately 10.13 g/100 g protein). Among the various types of essential amino acids present in tomato, leucine is present in the highest concentration, while methionine is the lowest. Among the nonessential amino acids, glutamic acid is the most common, while cysteine is the least.

### 3.6. Carotenoid Content

Tomato contains various types of carotenoids and is rich in lycopene and β-carotenoids (Table 6). Carotenoids are plant pigments that play crucial roles in protecting plants from photo-oxidative processes. They are natural antioxidants useful for combating cellular oxidative damage [82]. Recent studies have suggested that carotenoids play important roles in improving vision [83], are effective for preventing CVDs [84], protect against sperm health [85] and can prevent various types of cancer [7,86,87]. On the other hand, carotenoids such as lutein and zeaxanthin improve skin health [88]. Lycopene is a type of carotenoid found in tomato that is helpful in the prevention of liver, lung, prostate, breast, and colon cancers [70,89].

### 3.7. Sterol Content

Sterols are mainly found in plants, animals, and microorganisms. Plant sterols, also known as phytosterols, commonly occur as a mixture of β-sitosterol, campesterol and stigmasterol. Phytosterols play important roles in human health. Phytosterols block cholesterol absorption sites in the human intestine and reduce cholesterol absorption, leading to a reduction in low-density lipoprotein cholesterol (LDL-C) and prevention of CVD [95]. Research has also suggested that phytosterols exhibit an anticancer effect by inhibiting cancer cell growth, carcinogens, angiogenesis, and invasion of metastasis by promoting the apoptosis of cancerous cells [96,97]. Phytosterols also act as antioxidants to prevent oxidative stress [98]. Additional important functions of phytosterols are stimulation of the immune system and anti-inflammatory activities [99,100]. In fact, tomato is an excellent source of phytosterols. Approximately 1283 mg of phytosterols are present per kg of tomato. Among them, β-sitosterol and stigmasterol are the main ones (Table 7).

## 4. Antioxidant Properties and Bioactive Compounds in Tomato

Antioxidants are biomolecules that (1) prevent the oxidation of other molecules by inhibiting the initiation and elongation of the oxidizing chain reaction of ROS and (2) by inhibiting the proliferation of cells, free radical scavenging, the modulation of enzymatic activity via chelation of metallic ions and signal transduction pathways [101]. Antioxidants, which are bioactive reducing agents, prevent cellular damage caused by ROS, including superoxide anion radicals, hydroxyl radicals and hydrogen peroxide [102,103]. Antioxidants are considered as the first line of cellular defense used to minimize the harmful effects of free radicals by scavenging them while restoring the normal physiological system. Most medicinal plants are abundant sources of natural antioxidants, such as phenolic acid and flavonoids.

Tomatoes are consumed worldwide in various forms, either raw or processed and provide a significant amount of important antioxidants [104,105] including β-carotene, ascorbic acid, lycopene, tocopherol, phenolic acids, flavonoids, anthocyanins and other bioactive compounds (Table 8 and Table 9) [66,106,107,108]. The antioxidants present in tomato can delay, hinder, and prevent free radical oxidation, and thus forming stable radicals [109]. Generally, antioxidant compounds play important roles in the prevention of several human degenerative diseases, including CVDs, diabetes, cancer, neurological diseases, and aging, by minimizing oxidative stress caused by ROS [110,111].

Phenolic compounds present in tomato are considered as the primary antioxidant based on their ability to donate hydrogen atoms to reactive free radicals [104]. Carotenoid, ascorbic acid, lycopene, tocopherol, and anthocyanins are other important antioxidants in tomato. Carotenoids act as antioxidants by quenching singlet oxygen and peroxyl-radicals [112]. Normally, antioxidant activity is confirmed using 2,2-diphenyl-1-picrylhydrazyl (DPPH) assay, thiobarbituric acid reactive substance (TBARS) inhibition, ferric reducing power (FRAP) and ferrous ion chelating activities.

**Table 8 foods-10-00045-t008:** Antioxidant constituents in tomato (fresh weight).

Antioxidant Constituents	Units	Concentrations	Range	References
α-tocopherol	mg/100 g	0.701 ± 0.110	0.59–0.88	[3,5,49,50,61,65,90,113]
β-tocopherol	mg/100 g	0.030 ± 0.004	0.02–0.03	
γ-tocopherol	mg/100 g	0.810 ± 0.720	0.40–2.24	
δ-tocopherol	mg/100 g	0.020 ± 0.010	0.01–0.02	
Total tocopherol	mg/100 g	1.200 ± 0.150	1.02–1.44	
Vitamin C	mg/100 g	13.990 ± 3.73	9.03–23.80	
β-carotene	mg/100 g	0.420 ± 0.080	0.30–0.51	
Lycopene	mg/100 g	7.960 ± 1.780	5.02–9.49	
Phenolic acids	mg CIAE/g extract	25.500 ± 3.590	21.34–31.23	
Flavonoids	mg QE/g extract	4.230 ± 1.280	3.06–6.36	
Anthocyanins	mg ME/g extract	0.870 ± 0.470	0.23–1.36	

Concentrations are expressed as mean ± standard deviation. CIAE: chlorogenic acid equivalents. QE: quercetin equivalents. ME: malvidin 3-glucoside equivalents.

**Table 9 foods-10-00045-t009:** Bioactive compounds in tomato.

Name of Bioactive Compound	IUPAC Name	Molecular Formulas	Structure	Other Sources	References
Quercetin	3,5,7,3′,4′-pentahydroxyflavone	C_15_H_10_O_7_	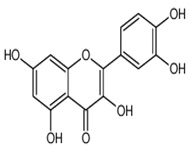	Onion family, chocolate, fortified wine, berry wine, grape wine, sparkling wine, cereals, herbs, fruit juices, tea infusions, spices, nuts, fruit	[3,5,13,44,92,104,114,115,116,117,118,119]
Kaempferol	3,5,7,4′-tetrahydroxyflavone	C_15_H_10_O_6_	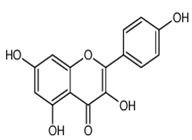	Pod vegetables, pulses, wine, fruit juices, spices, tea infusions, nuts, fruit	
Naringenin	5,7-Dihydroxy-2-(4-hydroxyphenyl)chroman-4-one	C_15_H_12_O_5_	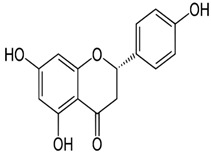	Herbs, fruits, vegetables, beans, drynaria	
Caffeic acid	3,4-dihydroxycinnamic acid	C_9_H_8_O_4_	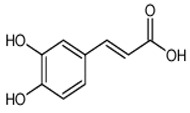	Grape wine, beer, fortified wine, cereal products, sparkling wine, cereals, tubers, berries, dried drupes, other dried fruits, citrus fruits, pomes, unknown coffee beverages, drupe juices, berry juices, pome juices, vegetable oils, herbs, other seasonings, tropical fruit juices, other types of seed oils, spices, fruit vegetables, cabbages, leafy vegetables, root vegetables	
Rutin	Rutoside; quercetin 3-rutinoside	C_27_H_30_O_16_	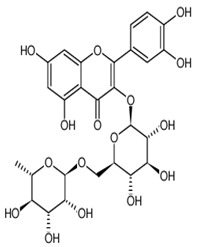	Invasive plant species (*Carpobrotusedulis)*	
Chlorogenic acid	3-caffeoylquinic and neochlorogenic acid	C_16_H_18_O_9_	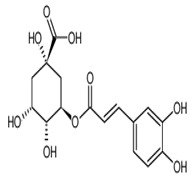	Grape wines, tea infusions, pomes, dried drupes, pome jams, tropical fruits, robusta coffee beverages, root vegetables, unknown coffee beverages, cabbages	
Ferulic acid	Transferulic acid; 4-hydroxy-3-methoxycinnamic acid	C_10_H_10_O_4_	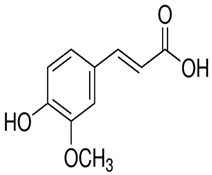	Cabbages, fruit vegetables, beer, fortified wine, sparkling wine, other dried fruits, cereals, soy based products, grape wine, cereal products, pome juices, chocolate, berries, citrus fruits, other fruits, pomes, berry juices, tropical fruit juices, fruit vegetable oils, other seed oils, herbs, nuts, beans and lentils	
*P*-coumaric acid	p-hydroxycinnamic acid	C_9_H_8_O_3_	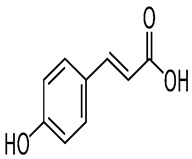	Sparkling wines, citrus fruits, beers, nut liquors, grape wines, fortified wines, dried drupes, other dried fruits, lentils, berries, other fruits, pomes, berry juices, pome juices, tropical fruit juices, fruit vegetable oils, other seed oils, herbs, other seasonings, spices, nuts, fruit vegetables	
Lycopene	lycopene, (7-cis,7’-cis 9-cis,9’-cis)-isomer	C_40_H_56_	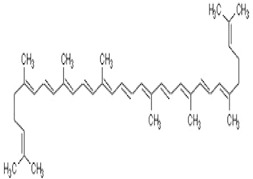	autumn olive, guac, guava, papaya, grapefruit, sea buckthorn, wolfberry, watermelon	
Resveratrol	3,5,4′-trihydroxy-*trans*-stilbene	C_14_H_12_O_3_	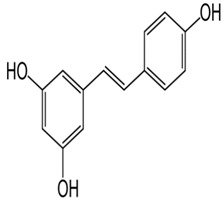	Grapes, wine, peanuts and soy	
Chrysin		C_15_H_10_O_4_	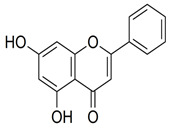	Fruit juices, especially citrus	
Epicatechin	(2R,3R)-2-(3,4-dihydroxyphenyl)-3,4-dihydro-2H-chromene-3,5,7-triol	C_15_H_14_O_6_	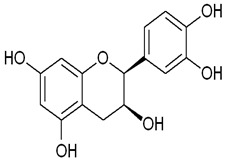	Tea, cocoa, grapes	
Catechin	(2*R*,3*S*)-2-(3,4-dihydroxyphenyl)-3,4-dihydro-2*H*-chromene-3,5,7-triol	C_15_H_14_O_6_	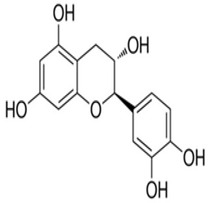	Tea, cocoa, grapes	
Luteolin	2-(3,4-dihydroxyphenyl)- 5,7-dihydroxy-4-chromenone	C_15_H_10_O_6_		Honey, fruit, vegetable oils, nuts, herbs, shoot vegetables, fruit, vegetables, lentils, carrots, olive oil	
Cinnamic acid	2-propenoic acid	C_9_H_8_O_2_	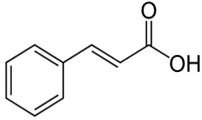	Propolis, fruit and vegetable oils, berries, citrus juices, fruit, vegetables, spices	
Phloretic acid	3-(4-hydroxyphenyl) propanoic acid	C_9_H_10_O_3_	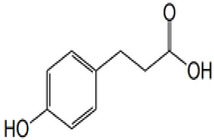	Dried grass	
Sinapic acid	3-(4-hydroxy-3,5-dimethoxyphenyl)prop-2-enoic acid	C_11_H_12_O_5_	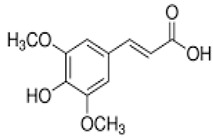	Black mustard seeds	
Vanillic acid	4-hydroxy-3-methoxybenzoic acid	C_8_H_8_O_4_	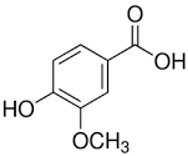	Rum, beer, brandy, wine, herbs, grapes, citrus fruit, whiskey	

## 5. Health Benefits of Tomato

The health benefits of tomatoes mainly associated with its rich supply of nutrients and secondary metabolites, including vitamins, minerals, essential fatty acids, carotenoids, antioxidants, and other bioactive compounds. In addition to its high amounts of vitamins A, C, E, K and B-complex [5,37], tomato is also a good source of important minerals as previously stated [3,47]. Additionally, tomato contains some dietary fiber, protein, essential amino acids, and a number of bioactive anti-oxidative organic compounds including lycopene, quercetin, kaempferol, naringenin, caffeic acid, rutin, resveratrol, catechin and luteolin, which contribute to the maintenance of good health [44]. Vitamins C and E are natural antioxidants that can prevent degenerative diseases caused by free radicals [120].

Lycopene is a natural antioxidant that can help combat different types of cancer, including prostate, breast, lung, stomach, colorectal, oral, esophageal, pancreatic, bladder, cervical and ovarian cancers [10,17]. Abundant amounts of minerals are responsible for maintenance of body’s physiological functions including blood pressure, blood clotting, nerve transmission, muscle contraction and energy production [56,121], while the vitamins help to maintain the health of the nervous system, facilitate blood cell production and enzymatic function [122].

The consumption of carotenoid-rich tomato has been reported to protect against vitamin A deficiency disorders and other chronic diseases including light-induced eye damage, the development of cataracts and age-related macular degeneration [123]. In fact, a high dietary intake of carotenoids (lutein and zeaxanthin) can prevent the risk of age-related macular degeneration, making tomato useful in ameliorating eye diseases [124]. Other important constituents present in tomato are phytosterols, which prevent intestinal cholesterol absorption by displacing it from the micelles, and thus stimulating its excretion, preventing CVDs, and ameliorating different types of cancer including colon, prostate, and breast cancers [1,125,126].

Oxidative stress is the main contributor to the development of chronic diseases in humans. ROS, including superoxide anion radicals, hydroxyl radicals and hydrogen peroxide, are highly reactive oxidant molecules that are endogenously induced by regular metabolic activity in the body, diet, and secondary lifestyle activities [127,128]. They react with cellular components (DNA, lipids, and proteins) to cause oxidative damage [129]. Antioxidants are super-protective agents that inactivate ROS and prevent oxidative damage [130]. Natural antioxidants such as vitamins C and E, different types of carotenoids and phenolic compounds including quercetin, kaempferol, caffeic acid, naringenin, chlorogenic acid, lutin, ferulic acid, lycopene, resveratrol, catechin and luteolin are present in tomato [6,92]. These bioactive compounds will protect endogenously produced reactive oxidant molecules and prevent oxidative damage. Therefore, they prevent the development of different types of cancer, diabetes and cardiovascular, eye, hypertension, inflammatory and neurodegenerative diseases [1,131,132,133,134].

In summary, the health benefits of tomato are associated with its rich supply of nutrients to the body, such as minerals, vitamins, proteins, essential amino acids, fatty acids, and other antioxidants. The consumption of tomato is associated with the relief of cancer, diabetes, CVDs, eye disease and constipation, with blood pressure reduction, improved blood circulation, improved body fluid balance, cholesterol reduction, detoxification of toxins, reduction in inflammation, prevention of premature aging and improvement of digestive function (Figure 1).

## 6. Effects of Bioactive Compounds of Tomato on Some Human Degenerative Diseases

### 6.1. Tomato in CVDs

Generally, CVD is a category of disease that affects the blood vessels and the heart [135]. Common predisposing factors include hypertension, gender, age, hypercholesterolemia, obesity, diabetes, and unhealthy lifestyle, such as minimal physical activity, smoking, the consumption of a high fat diet and excessive alcohol [136,137]. The bioactive compounds in tomato not only reduce the risk but also prevent or ameliorate CVDs [138] (Figure 2). The antiplatelet aggregation effects of tomato and tomato products support the prevention of CVD disorders [139]. For example, lycopene can improve endothelial function among patients who suffer from CVD [140]. Lycopene functions as a crucial hypolipidemic and antioxidant agent and inhibits factors that play important prothrombosis and pro-inflammatory roles, and thus improves CVDs [141]. Additionally, it is hypothesized that lycopene can increase LDL degradation and reduce cholesterol synthesis. It has been reported that both the thickness of the intima wall and myocardial infarction (MI) can be minimized by high lycopene intake [142]. As a radical scavenger, lycopene mops up singlet oxygen along with other active free radicals, and thus protects against vascular cell damage, which contributes to CVD [141]. Lycopene also shows antiplatelet and antithrombotic activities by hindering phospholipase C activation, which inhibits the breakdown of phosphoinositol and the formation of thromboxane B2. Subsequently, the mobilization of intracellular calcium, which is beneficial in CVDs, is impeded. In addition, the cyclic guanosine monophosphate (cGMP) /nitrate formation in the platelets activated by lycopene also constrains platelet aggregation [143].

Quercetin is a potential chelator of metal ions, constraining xanthine oxidase while reducing lipid peroxidation and scavenging free radicals, which ultimately help to reduce the risk of CVD [144,145]. Furthermore, it reduces the levels of oxidized LDL in the plasma, since LDL oxidation is stimulated by macrophages or myeloperoxidase, inhibits the generation of hyperoxide and reduces systolic blood pressure in obese patients [146,147,148]. Quercetin also decreases the expression of endothelin-1 messenger ribonucleic acid (mRNA), which stimulates the dilation of coronary vessels and improves endothelial function [149]. Another study showed that quercetin regulates the expression of p47^phox^ or neutrophil cytosol factor 1 and therefore reduces the levels of superoxide anion (O_2_^−^), as mediated by nicotinamide adenine dinucleotide phosphate oxidase (NADPH oxidase), ultimately preventing endothelial dysfunction in hypertension [150].

Caffeic acid, which can increase the plasma level of vitamin E, scavenge free radicals, and reduce oxidative stress, demonstrates its effectiveness in MI [151,152]. It is also reported to be a potential antihypertensive agent as well [153]. Kaempferol effectively reduces oxidative stress, which is effective in CVD. On the other hand, naringenin, which is efficacious against interstitial fibrosis and cardiac hypertrophy, can improve the function of the left ventricle in mice with induced pressure overload. It also downregulates the activation of the c-Jun N-terminal kinases (JNK), extracellular signal–regulated kinases (ERK) and phosphatidylinositol-4,5-bisphosphate 3-kinase (PI3K)/protein kinase B (AKt) signaling pathways, which may offer some cardioprotective effects [154].

Lutein reduces carotid artery intima-media thickness, especially in patients with subclinical atherosclerosis [155]. Moreover, the use of dietary lutein has been reported to reduce atherosclerotic lesions, especially in combination with polyunsaturated fatty acids (PUFAs) [156]. Luteolin confers cardioprotective mechanisms against ischemia/reperfusion (I/R) injury by increasing pro-apoptotic molecules, Bcl2-associated agonist of cell death (BAD) phosphorylation and manganese superoxide dismutase (MnSOD) activity, which helps to inhibit the mitochondrial permeability transition pore (mPTP), stimulate the PI3K/AKt as well as myocardial endothelial nitric oxide synthase (eNOS) pathways, upregulate leukemic inhibitory factor (LIF) and anti-apoptotic proteins such as fibroblast growth factor receptor 2 (FGFR2) expression and decrease the ratio of Bax to Bcl-2, particularly in diabetes [157,158]. Moreover, luteolin inhibits thrombin activity, fibrin polymer formation and thrombosis, which is induced by oxidative stress and binds with thromboxane A2 receptor, which can hinder platelet aggregation, and thus confers some antithrombotic effect [159,160].

Vanillic acid reduces hypertension and infarct size in I/R, improves ventricular function and shows antioxidant mechanisms that can confer some cardioprotective effect [161,162,163]. Chlorogenic acid demonstrates antihypertensive, antiplatelet, antithrombotic and antioxidant activities involving A2A receptor and NF-κB and the adenylate cyclase/cyclic adenosine monophosphate/protein kinase A (adenylatecyclase/cAMP/PKA) pathways, which can improve CVDs [164,165,166]. On the other hand, ferulic acid improves the structure as well as the function of blood vessels and the heart [167]. Additionally, p-coumaric acid exhibits a preventive role in cardiac hypertrophy and MI by stimulating anti-hypertrophic and radical scavenging activities that constrain lysosomal dysfunction and ameliorate the levels of lysosomal enzymes [168,169].

On the other hand, chrysin has an anti-atherogenic activity and can ameliorate MI by activating peroxisome proliferator-activated receptor gamma (PPAR-ɣ) and inhibiting oxidative stress as well as inflammation, mediated by the advance glycation end product [170,171]. Catechin regulates lipid and blood lipid metabolism, reduces, and regulates blood pressure, protects vascular endothelial cells, reduces the cell proliferation of vascular smooth muscle, suppresses platelet adhesion, inhibits thrombogenesis and increases vascular integrity, thereby reducing the risk for CVDs [172,173]. Epicatechin can decrease platelet-induced endothelial activation along with catechin [174]. Cinnamic acid ameliorates myocardial ischemia due to its anti-inflammatory and anti-oxidative properties [175]. On the other hand, sinapic acid has membrane-stabilizing and free radical scavenging properties by which it can inhibit fibrosis and lysosomal dysfunction in heart diseases [176,177]. The protective effects of resveratrol occur by the upregulation of AMP-activated protein kinase (AMPK) and sirtuin (SITR1) and the activation of endogenous antioxidant enzymes against cardiovascular complications. It also confers some lipid-reducing, antiplatelet and anti-inflammatory properties that are beneficial in CVDs [178].

### 6.2. Tomato in Diabetes

Some bioactive compounds present in tomato are effective in diabetes (Figure 2). For instance, lycopene has been reported to exert hypoglycemic effects by increasing serum insulin levels and lowering glucose levels in diabetic animals as induced by streptozotocin (STZ) [179,180]. Lycopene reduces angiotensin converting enzyme (ACE) activity, the level of which can indicate diabetes or complications related to diabetes [181]. In addition, it has been reported to improve renal function and exhibit a defensive effect against diabetic nephropathy by regulating connecting tissue growth factor and p-Akt, reducing malondialdehyde levels and enhancing antioxidant activities [182]. It has also been reported that oocyte maturation, follicular growth and protection of ovaries are promoted by lycopene in diabetic conditions [183].

Quercetin stimulates glucose uptake by regulating mitogen-activated protein kinase (MAPK) insulin-dependent mechanisms, improving renal function, inhibiting the overexpression of transforming growth factor beta 1 (TGF-β1) and connective tissue growth factor (CTGF), hindering polyol accumulation as well as blocking aldose reductase, and thus reducing joint pain, irritation, and numbness, which are common symptoms observed in diabetes [184]. Quercetin elevates transmembrane potential and membrane fluidity and confers anti-inflammatory activity on immune and endothelial cells, which is beneficial especially in the late stage of diabetes [185]. Due to its antiplatelet activity, quercetin delays thrombus formation and plays a crucial role as an antioxidant by decreasing the generation of lipid hydroperoxides and increasing the activity of glutathione peroxidase in diabetes [186,187]. It also regulates NF-κB signaling and the mitochondrial pathway, significantly preventing the death of β cells [188].

On the other hand, kaempferol minimizes α-glucosidase activity, increases antioxidant activity, reduces lipid peroxidation level, protects β cell function and improves insulin sensitivity of the periphery, therefore exhibiting antidiabetic properties [189,190,191]. Kaempferol also downregulates I kappa B kinase (IKK), and thus inhibits the NF-κB pathway, which reduces inflammatory lesions in hepatic cells and improves insulin signaling [192]. Kaempferol also restores membrane-bound ATPase, which is normally affected in diabetes [193].

Naringenin, which has the potential to diminish nephropathy and improve endothelial complications along with lipid and glucose metabolisms in diabetes, shows antifibrotic, anti-oxidative and anti-inflammatory activities [194,195]. Interestingly, naringenin also inhibits increased cholinesterase (ChE) activity, which contributes to memory dysfunction [196]. Therefore, it is another important component present in tomato.

Caffeic acid has been reported to confer some anti-inflammatory and antiglycemic activities in diabetic kidney disease [197]. Chlorogenic acid reduces plasma glucose levels in fasting conditions and during late diabetes, where it lowers glycosylated hemoglobin (HbA1c). Additionally, by modulating the signaling pathway of the adiponectin receptor, chlorogenic acid improves diabetic kidney fibrosis [198]. It also prevents neurological complications and retinopathy accelerated by diabetes by improving memory, inhibiting TBARS production, reducing anxiety and reducing vascular hyperpermeability [199,200].

In pregnant women affected by gestational diabetes, lutein intake has been reported to reduce neonatal oxidative stress during birth [201]. Moreover, lutein, which is another important component present in tomato, prevents cataract progression and preserves the composition of free fatty acids, which is abnormal in diabetic conditions [202,203]. Luteolin can inhibit α-glucosidase action and improve insulin resistance [204,205], which is important in both diabetic encephalopathy and neuropathy [206,207].

Vanillic acid reduces blood pressure, plasma glucose and insulin by activating antioxidants, thereby reducing oxidative stress [208]. Ferulic acid reduces oxidative stress, and thus prevents testicular damage and downregulates both apoptosis and pro-inflammatory cytokine expression in diabetes [209,210]. Another important substance is p-coumaric acid, which exerts glucose-lowering activity by activating pancreatic glucose transporter 2 (GLUT 2), regulating lipid and glucose metabolisms and exhibiting some antidiabetic effects [211]. Moreover, p-coumaric acid shows anti-apoptotic, anti-inflammatory and antioxidant activities that counter hippocampal neurodegeneration in diabetes [212].

Cinnamic acid and some of its derivatives have the potential to be applied in the treatment of diabetes because of their many useful properties including (1) increased glucose uptake, adiponectin secretion, insulin secretion and hepatic glycolysis; (2) improved functionality of pancreatic β cells; (3) reduced adipogenesis; (4) improved hepatic gluconeogenesis, protein glycation, insulin fibrillation and intestinal glucose absorption and (5) decreased activity of certain enzymes, including α-glucosidase, dipeptidyl peptidase-4, pancreatic α-amylase and protein tyrosine phosphatase 1B [213].

Sinapic acid improves hyperglycemia and maximizes glucose utilization by regulating the signals of phospholipase C (PLC) and protein kinase C (PKC) in diabetes [214]. Catechin activates endothelial phosphoinositide (PI3K) signaling and consequently activates eNOS, resulting in the generation of nitrous oxide against vascular endothelial dysfunction (VED) in diabetes. In cases of vascular endothelial abnormalities (VEA), catechin confers a protective effect by reducing high glucose, lipid peroxidation and oxidative stress [215]. In addition, catechin protects against diabetic nephropathy by constraining the formation of advanced glycated end products and blocking inflammatory signaling pathways since catechin can trap the metabolite methylglyoxal [216].

Epicatechin, another important constituent of tomato, reduces insulin resistance, increases insulin sensitivity, and reduces oxidative stress [217]. It also improves pancreatic insulitis and islet mass as well as muscle function [218,219]. Chrysin hinders the activity of α-glucosidase, reduces oxidative stress, and generates moderate amounts of nitric oxide to prevent diabetes-related complications [220,221]. In addition to its antioxidant properties, chrysin has anti-inflammatory properties and the ability to regulate the apoptotic cascade by which it improves diabetic-associated cognitive deficits (DACD) and diabetic nephropathy [222,223]. Resveratrol is another important component present in tomato. In addition to having a significant effect on different signaling and metabolic pathways that can improve diabetes, it enhances mitochondrial biogenesis and reduces mitochondrial damage, oxidative damage, inflammation, lipid accumulation, liver steatosis and improves the action of insulin [224].

### 6.3. Tomato against Cancer

Many recent studies have suggested that regular intake of fruits and vegetables that are high in antioxidants prevents the progression of cancer cells [14,225]. There are many important bioactive phytochemicals present in fruits and vegetables that are responsible for cancer prevention. For example, bioactive phytochemicals can prevent cancer via various mechanisms including (1) inhibition of cancer cell proliferation [226], (2) prevention of oxidative stress via antioxidant effects [227], (3) alteration of cancer cell signaling pathways [228,229], (4) modification of enzymatic activity [230], (5) inhibition of oxidative DNA damage [231], (6) increasing the expression of phase II enzymes [232], and (7) inducing apoptosis of cancer cells [233,234] (Figure 2).

Many epidemiological studies have suggested that dietary intake of tomato can ameliorate cancer [13,70]. In fact, the excessive consumption of tomato is reported to confer some preventive effects against gastrointestinal tract cancer when compared to a control [235]. A research conducted by Colditz et al. confirmed that the intake of tomato and other types of vegetables is important for cancer prevention. In their research, from the 1271 elderly individuals investigated, 42 cancer patients died. However, those with higher intakes of tomato had a better prognosis [236]. Studies from different countries have reported that colon cancer is inversely correlated with a high tomato consumption [237,238]. Moreover, a case-control study from China and Italy indicated that approximately 60% of colon and rectal cancer patients showed some improvements following the intake of high amounts of tomato [235,239,240].

Lycopene, a major carotenoid present in tomato, has antioxidant properties and can prevent prostate cancer by blocking ROS generation, scavenging free radicals, and protecting both cell membrane and DNA from oxidative damage [10,17,241]. In another study, approximately 83% of prostate cancer was ameliorated in patients who received the highest amount of lycopene (0.40 µm/L) in comparison with lower amounts (0.18 µm/L) [242]. Siler et al. investigated the protective effects of lycopene and vitamin E (two important components present in tomatoes) on the growth of prostate tumors in a Dunning MAT-LyLu rat model [243], where rapidly growing tumor cells were injected into the ventral prostate of the experimental rats. Both lycopene- and vitamin E-treated experimental rats showed a significant reduction in tumor size and necrosis when compared to untreated rats. Moreover, molecular analysis of the tumor tissues indicated that vitamin E reduces androgen signaling, while lycopene downregulates the expression of 5-α reductase 1, interleukin 6 and insulin-like growth factor-1.

Quercetin confers both antioxidant and anticancer properties that prevent the proliferation of prostate cancer cells by increasing the levels of p^21^, Bax and caspase-3, and thus preventing the expression of the cell cycle regulator cdc2/cdk-1, cyclin B1, phosphorylated p^RB^ and apoptosis markers Bcl-2 and Bcl-X_L_ [244]. In another study, quercetin showed anticancer properties by inhibiting the P13k-Akt/PKB pathway [245]. Quercetin also protects against colon cancer by inhibiting β-catenin/Tcf signaling in SW480 colon cancer cell lines and reducing β-catenin/Tcf transcription activity [246]. Luteolin, which is also present in tomato, is an important polyphenol that exhibits anticancer properties by inhibiting rat aortic vascular smooth muscle cell proliferation and DNA synthesis as induced by platelet-derived growth factor-BB (PDGF-BB) via blockage of the phosphorylation of PDGF-BB receptor [247]. Apigenin, another important flavonoid in tomato, also has some anticancer properties since it inhibits pancreatic cancer cell proliferation by arresting the G2/M cell cycle, and thus decreases the concentration of cyclin A, cyclin B and the phosphorylated forms of cdc2 as well as cdc25 [248].

## 7. Limitations and Future Prospects

The nutritional composition of tomato depends on the maturation of fruits, ripening time, geographical location, tomato variety, freshness and whether they are tomato-based food products. The nutritional composition will also differ depending on the conditions of sample preparation, collection, and tomato parts. Nevertheless, to date, there are limited data on the nutritional contents of tomato peel and pulp for a good comparison to be made. Therefore, in this review, only the nutritional composition of fresh matured tomatoes is discussed. Further study needs to investigate the bioactive compounds present and investigate their impact on human health.

Additionally, although some research works have been published on tomato processing by-products containing many bioactive nutrients including lycopene, beta-carotene, amino acids, proteins, lipids, and dietary fibers [249,250], the bioavailability, safety, and pharmacological toxicity have not been investigated. Another study Salehi et al. demonstrated that tomato consumption, in addition to giving some beneficial effects, is also associated with some health risks including renal problems, irritable bowel syndrome (IBS), migration, allergy, body aches, arthritis, and urinary problems [1]. Therefore, research should also focus on the possible health risks that may arise due to tomato consumption and the amount consumed that leads to detrimental effects. Finally, there is a need to conduct randomized clinical trials that can assess the effects of long-term tomato consumption on its various health benefits.

## 8. Conclusions

Tomatoes are vegetables/fruits that contain significant amounts of dietary nutrients, including dietary fiber, reducing sugars, vitamins, minerals, protein, essential fatty acids, phytosterols and carotenoids. The nutritive elements play an important role in bodily function and are beneficial in ameliorating chronic diseases. Tomatoes are also rich with health promoting bioactive phytochemicals, such as phenolic compounds including lycopene, quercetin, kaempferol, naringenin, caffeic acid and lutein. The bioactive constituents show antioxidant, antiproliferative, antidiabetic, anti-inflammatory, and other health-promoting activities, indicating the vast potential of tomato in preventing and/or ameliorating several chronic degenerative diseases. Lycopene and β-carotene are two main active ingredients in tomato that have strong antioxidant properties, which are linked with many health benefits, including cancer and heart diseases. They also participate in preventing the development of cataracts. Additionally, the water content and dietary fibers help the body in terms of hydration, bowel movements, reducing constipation, improving obesity through weight loss, and preventing colon cancer. All immune stimulating activities of tomatoes make them active ingredients for the development of functional foods. Thus, tomato is an excellent source of dietary nutrients and is useful in disease prevention.

## Figures and Tables

**Figure 1 foods-10-00045-f001:**
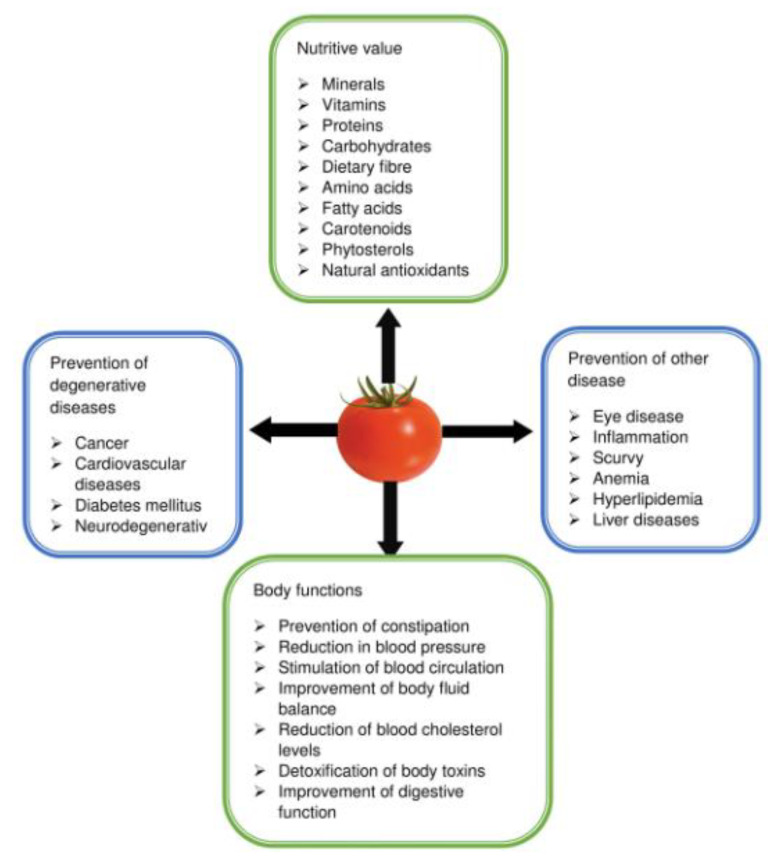
Summary of health benefits of tomato.

**Figure 2 foods-10-00045-f002:**
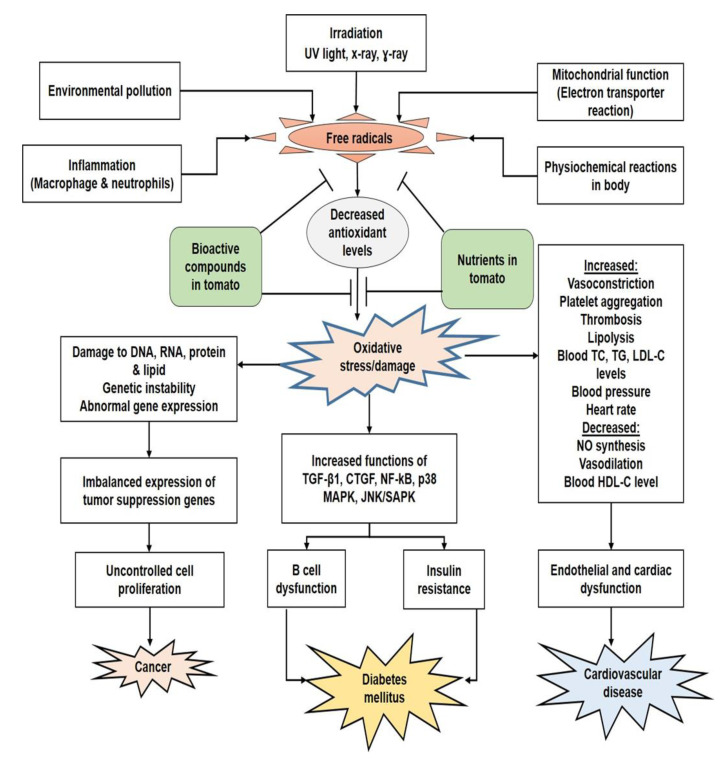
Summary of the effects of nutrients and bioactive compounds in tomato on cancer, diabetes mellitus and cardiovascular disease (CVD). UV: ultraviolet; DNA: deoxyribonucleic acid; RNA: ribonucleic acid; TGF-β1: transforming growth factor beta 1; CTGF: connective tissue growth factor; NF-kB: nuclear factor kappa-light-chain enhancer of activated B cells; p38 MAPK: p38 mitogen-activated protein kinase; JNK/SAPK: c-Jun N-terminal kinases/stress-activated protein kinases; TC: total cholesterol; TG: triglyceride; LDL-C: low-density lipoprotein cholesterol; NO: nitric oxide; HDL-C: high-density lipoprotein cholesterol.

**Table 1 foods-10-00045-t001:** Proximate composition of tomato.

Parameters	Values	Range	USDA Values	References
Energy (kcal/100 g)	26.60 ± 5.56	18.00–34.67	22 Kcal/100 g	[3,4,5,40,41,42,43,44,45,46,47,48,49,50,51,52]
Ash (%)	0.60 ± 0.09	0.50–0.74	0.31 g	
Moisture (g/100 g)	93.76 ± 1.75	90.063–96.17	94.7 g/100 g	
Total protein (g/100 g)	0.56 ± 0.19	0.40–0.88	0.7 g/100 g	
Lipid (g/100 g)	0.13 ± 0.06	0.03–0.20	0.42 g/100 g	
Carbohydrates (g/100 g)	5.96 ± 1.37	3.92–8.00	3.84 g/100 g	
Total sugar (g/100 g)	4.41 ± 1.69	2.49–6.62	-	
pH	4.38 ± 0.16	4.13–4.57	-	
Acidity (%)	0.48 ± 0.07	0.39–0.55	-	
Reducing sugar (%)	35.84 ± 4.57	30.03–41.21	-	
Fructose (%)	2.88 ± 0.49	1.15–3.42	-	
Glucose (%)	2.45 ± 0.48	1.74–3.18	-	
Sucrose (%)	0.02 ± 0.05	0.01–0.02	-	
Total fiber (g/100 g)	1.54 ± 0.43	0.70–1.92	1 g/100g	

Values are expressed as mean ± standard deviation.

**Table 2 foods-10-00045-t002:** Mineral contents in tomato peel fiber.

Elements	Units	Concentrations	Range	References
Sodium (Na)	mg/100 g	70.38 ± 12.20	56.90–80.65	[3,4,41,42,44,45,47,51,61,62,63]
Potassium (K)	mg/100 g	403.02 ± 254.41	16.63–1097.00	
Calcium (Ca)	mg/100 g	105.21 ± 22.76	48.47–162.07	
Magnesium (Mg)	mg/100 g	172.58 ± 58.92	76.87–265.93	
Phosphorus (P)	mg/100 g	300.99 ± 32.12	173.00–379.31	
Chlorine (Cl)	μg/100 g	517.24 ± 0.00	517.24	
Boron (B)	μg/g	36.83 ± 3.27	25.84–48.59	
Nickel (Ni)	mg/100 g	0.66 ± 0.00	0.66	
Nitrate (NO_3_^-^)	mg/100 g	274.37 ± 156.75	86.21–459.00	
Iron (Fe)	mg/100 g	4.55 ± 2.18	1.50–6.45	
Zinc (Zn)	mg/100 g	2.48 ± 1.05	0.17–3.17	
Cobalt (Co)	mg/100 g	19.66 ± 9.66	10.00 -29.31	
Copper (Cu)	mg/100 g	0.67 ± 0.15	0.06–1.10	
Manganese (Mn)	mg/100 g	0.60 ± 0.12	0.11–1.88	
Chromium (Cr)	μg/100 g	193.80 ± 133.80	60.00–327.59	
Iodine (I)	mg/100 g	2.65 ± 1.44	0.18–3.97	
Fluorine (F)	μg/100 g	413.79 ± 0.00	413.79	
Aluminum (Al)	μg/100 g	1241.38 ± 0.00	1241.38	
Silicon (Si)	μg/100 g	46.55 ± 0.00	46.55	
Selenium (Se)	μg/100 g	13.45 ± 3.45	10.00–16.90	
Lead (Pb)	μg/ g	1.21 ± 0.06	1.15–1.27	
Cadmium (Cd)	μg/ g	0.17 ± 0.06	0.11–0.22	
Arsenic (As)	μg/ g	0.20 ± 0.005	0.19–0.20	

Concentrations are expressed as mean ± standard deviation.

**Table 3 foods-10-00045-t003:** Vitamin contents in tomato.

Vitamins	Units	Concentrations	Range	USDA Values	References
Vitamin A	IU/100 g	614.44 ± 248.18	267.33–833.00	24 µg/100 g	[4,46,61,62,65,69,73,74,75,76]
Vitamin E	mg/100 g	0.37 ± 0.15	0.17–0.62	-	
Vitamin K	μg/100 g	98.28 ± 0.00	98.28	-	
Vitamin C	mg/100 g	13.99 ± 3.73	9.03–23.80	17.8 mg/100 g	
Thiamine	mg/100 g	0.66 ± 0.44	0.04–0.98	0.056 mg/100 g	
Riboflavin	mg/100 g	0.48 ± 0.34	0.02–0.81	<0.1 mg/100 g	
Pantothenic Acid	mg/100 g	4.93 ± 0.41	4.52–5.34	-	
Vitamin B6	mg/100 g	1.51 ± 0.22	1.29–1.72	0.079 mg/100 g	
Biotin	μg/100 g	68.97 ± 0.00	68.97	0.469 µg/100 g	
Folate	μg/100 g	14.00 ± 1.00	13.00–15.00	10 µg/100 g	

Concentrations are expressed as mean ± standard deviation.

**Table 4 foods-10-00045-t004:** Fatty acid contents (percentage in total fatty acid content) in tomato (dry weight).

Fatty Acids	Concentrations (g/100 g in Total Fatty Acid)	Range	References
Myristic acid	0.56 ± 0.22	0.32–0.93	[5,41,47,49,62]
Palmitic acid	18.07 ± 2.90	12.40–22.50	
Stearic acid	4.81 ± 1.50	2.80–6.84	
Palmitoleic acid	0.25 ± 0.10	0.03–0.32	
Oleic acid	14.24 ± 3.50	9.00–19.14	
Linoleic acid	49.40 ± 4.16	46.33–54.10	
Linolenic acid	10.17 ± 4.46	4.26–15.53	
Caproic acid	0.03 ± 0.02	0.01–0.05	
Caprylic acid	0.06 ± 0.04	0.02–0.10	
Capric acid	0.04 ± 0.03	0.01–0.07	
Heptadecanoic acid	0.26 ± 0.05	0.18–0.13	
Lauric acid	0.09 ± 0.05	0.04–0.15	
Pentadecanoic acid	0.12 ± 0.03	0.08–0.15	
Arachidic acid	0.88 ± 0.24	0.61–1.26	
Eicosadienoic acid	0.04 ± 0.02	0.02–0.06	
Arachidonic acid	0.04 ± 0.02	0.01–0.06	
Eicosapentaenoic acid	0.05 ± 0.01	0.03–0.06	
Erucic acid	0.02 ± 0.01	0.01–0.03	
Docosadienoic acid	0.07 ± 0.03	0.03–0.10	
Behenic acid	0.59 ± 0.19	0.31–0.82	
Tricosanoic acid	0.68 ± 0.54	0.16–1.52	
Lignoceric acid	0.74 ± 0.20	0.45–1.01	
Saturated fatty acid	27.40 ± 3.74	22.37–33.22	
Monounsaturated fatty acid	13.80 ± 2.42	11.00–17.66	
Polyunsaturated fatty acid	57.55 ± 23.51	55.78–58.63	
Vaccenic acid	0.53 ± 0.05	0.50–0.60	
Eicosanoic acid	0.10 ± 0.03	0.05–0.12	

Concentrations are expressed as mean ± standard deviation.

**Table 5 foods-10-00045-t005:** Amino acid contents of tomato.

Amino Acids	Concentrations (g/100 g Protein)	Range	References
Threonine *	1.37 ± 0.97	0.40–2.34	[3,41,62,80,81]
Valine *	2.49 ± 2.09	0.40–2.49	
Methionine *	0.57 ± 0.45	0.12–1.02	
Isoleucine *	2.13 ± 1.73	0.40–3.86	
Leucine *	2.80 ± 2.28	0.52–5.07	
Phenylalanine *	1.77 ± 1.36	0.41–13.12	
Histidine *	1.93 ± 1.71	0.22–3.64	
Lysine *	2.45 ± 1.95	0.50–4.40	
Arginine *	2.33 ± 2.02	0.31–4.34	
Aspartic Acid **	1.40 ± 0.70	0.70–2.09	
Serine **	1.78 ± 1.30	0.48–3.08	
Glutamic Acid **	10.13 ± 4.44	5.69–14.56	
Proline **	1.53 ± 1.25	0.28–2.78	
Glycine **	2.30 ± 1.99	0.31–4.29	
Alanine **	2.74 ± 2.29	0.45–5.02	
Cystine **	0.21 ± 0.19	0.02–0.39	
Tyrosine **	1.82 ± 1.61	0.21–3.42	

Concentrations are expressed as mean ± standard deviation. * denotes essential amino acids. ** denotes nonessential amino acids.

**Table 6 foods-10-00045-t006:** Carotenoid contents in tomato.

Carotenoids	Units	Concentrations	Range	USDA Values	References
β-carotene	mg/100 g FW	0.420 ± 0.080	0.30–0.51	276 µg/100 g	[5,7,41,46,47,74,90,91,92,93,94]
Lycopene	mg/100 g FW	7.960 ± 1.780	5.02–9.49	2860 µg/100 g	
Lutein + zeaxanthin	μg/100 g FW	60.67 ± 43.86	18.07–123.00	56 µg/100 g	
Phytoene	μg/100 g FW	668.33 ± 361.95	430.00–1860.00	-	
Phytofluene	μg/100 g FW	500.00 ± 100.49	390.00–820.00	-	
All trans-lutein	mg/kg DW	5.00 ± 0.82	4.00–6.00	-	
All trans-β carotene	mg/kg DW	29.25 ± 27.26	4.00–75.00	-	
9-cis-β carotene	mg/kg DW	6.50 ± 2.29	3.00–9.00	-	

Concentrations are expressed as mean ± standard deviation. FW = fresh weight, DW = dry weight.

**Table 7 foods-10-00045-t007:** Sterol contents in tomato (mg/kg dry wt).

Sterols	Concentrations (mg/kg)	Range	References
Campesterol	147.50 ± 31.13	100.00–180.00	[5,92]
Stigmasterol	387.50 ± 88.71	260.00–510.00	
Stigmastanol	28.25 ± 10.92	10.00–38.00	
β-sitosterol	720.00 ± 175.64	520.00–1000.00	
Δ5-Avenasterol	62.30 ± 2.21	10.00–65.87	
Cholestanol	9.70 ± 1.80	2.10–11.54	
Cholest-7-en-3-ol	3.60 ± 0.13	0.42–4.40	
Cholesterol	41.90 ± 2.10	8.40–43.45	
Lanost-8-en-3-β-ol	52.40 ± 6.80	4.50–60.65	
24-Oxocholesterol	67.50 ± 3.20	14.20–70.69	
Total sterol	1283.25 ± 239.39	918.00–1570.00	

Concentrations are expressed as mean ± standard deviation.

## Data Availability

Not applicable.

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
