# Peer review of "Nutritional Composition and Bioactive Compounds in Tomatoes and Their Impact on Human Health and Disease: A Review"

_foods, 2020, doi:10.3390/foods10010045_

Round 1

Reviewer 1 Report

The effects of tomato consumption and its components in human health and disease, is a relevant and interesting topic. Authors performed an extensive review. Unfortunately, the review is not timely. Just about 4% of the references are from the last 3 years, about 19% are from the past century, and around 48% are form more than years ago. Authors should perform a deep actualization of the literature referenced to add some value to the review, which topic has been reviewed several times, and discuss the state of the art of this relevant issue. In the limitations section, authors should discuss the need of randomized trials assessing the effects of long term tomato consumption on hard endpoints.

Reviewer 2 Report

Manuscript ID: foods-1030124

Title:  Nutritional Composition and Bioactive Compounds in Tomatoes and their Impact on Human Health and Disease: A Review

Major comments:

Introduction section need significant revision, it is very short and please highlight the knowledge gap.

Please replace the chemical structure of Lycopene in “Table 9. Bioactive compounds in tomato”.

All the tables should be according format of MDPI-Foods.

Conclusion section need to be revised, please provide the brief conclusion at least 10-15 lines with key highlights of this review.

Round 2

Reviewer 1 Report

The authors have significantly improved the manuscript.